# Theoretical and Experimental Flow Characteristics of a Large-Scale Annular Channel in Terms of Deformation Gradient, Eccentricity, and Water Compressibility

Shendan Zhao, Yinshui Liu *, Defa Wu, Chuanmin Wang and Zhenyao Wang

State Key Laboratory of Digital Manufacturing Equipment and Technology, School of Mechanical Science and Engineering, Huazhong University of Science and Technology, 1037 Luoyu Road, Wuhan 430074, China; zhaoshendan2021@163.com (S.Z.)
* Correspondence: liuwater@hust.edu.cn

**Abstract:** Hydraulic water plunger pumps have come to be widely used in coal mining, seawater desalination, and oil exploitation due to their high output pressure and large flow characteristics. In a high-pressure large-flow plunger pump, the leakage of the annular channel of the plunger pair is an essential factor affecting volume efficiency. The axial pressure gradient exists in the fluid inside the annular channel, resulting in the plunger and plunger sleeve forming similar funnel-like shapes. Moreover, the characteristics of large diameter, high working pressure, and low fluid viscosity of the plunger pump will lead to the complicated flow of the annular channel. The influence of eccentricity and structural deformation on leakage is difficult to evaluate. Therefore, considering the deformation gradient and eccentricity of the plunger pair and the compressibility of the water, the deformation equations and leakage equations of the annular channel under the laminar and turbulent flow state are derived in this study. The eccentricity and leakage of the annular channel under different pressure conditions are measured using a built sealing test bench. It is proved that the discrepancy between the calculated model and the experimental results is less than 6% under different pressures, which effectively predicts the sealing performance of plunger pumps. The results show that under the laminar flow condition, the effects of eccentricity, structural deformation, and medium compressibility on leakage are 148%, 4.92%, and 0.92%, respectively. In turbulent conditions, they were 31%, 2.84%, and 1.19%, respectively. Besides, the reasonable material pairing of the plunger friction pair can reduce the variation of leakage due to structural deformation.

**Keywords:** large-scale annular channel; elastic deformation; fluid compressibility; gap seal test bench

## 1. Introduction

As water hydraulic technology has developed, hydraulic radial plunger pumps have come to be widely used in coal mining, seawater desalination, and oil exploitation due to their high output pressure and large flow characteristics [1–3]. However, the *pv* value of the actual work exceeds the capacity limit of the packing, V-ring, and other sealing rings due to the high pressure and large scale of the plunger pump, resulting in a sharp decline in the service life of the dynamic seals. Compared to the traditional contact seal, the friction pair of the gap seal is separated by a water film, which can reduce the friction coefficient and has outstanding advantages in the field among high-speed dynamic seals.

Annular channel flow is a classical hydrodynamic problem [4]. Due to the throttling effect of the gap, the fluid pressure changes along the axial direction. Under the high working pressure, not only is the structure of the plunger pair deformed like a funnel but the fluid density in the plunger cavity is also changed. Furthermore, the plunger and the plunger sleeve are often eccentric due to the biased load and machining errors in the actual work. These factors may affect the leakage of the plunger pair together. Therefore, the

deformation and leakage mechanism of the annular channel must be considered to improve the accuracy of the calculation model.

Given that the gap in the plunger pair is much smaller than its diameter, the flow at the gap fit of the plunger pair forms a Poiseuille–Couette flow between two plates. Trutnovsky [5] derived a calculation formula for the leakage of an annular channel when a fluid is in the state of laminar and turbulent flow. Yang [6] studied the pressure distribution of an annular channel and the law of leakage during the reciprocating movement of the large-scale plunger. Jiang et al. [7] examined the working efficiency and reliability of a plunger friction pair by calculating the shape and pressure distribution of the pair in a radial plunger pump. Deng et al. [8] proposed a laddered piston assembly with a seal gap. The effects of the seal length, shaft speed and seal gap on the sealing efficiency were studied by considering the gas characteristics in the compression chamber and real-time variation in piston movement in a thermal process. Kakoi [9] adopted the pressure gradient coordinate system and proposed a non-Newtonian isothermal flow point contact elastic flow lubrication analysis formula. Kyritsi-Yiallourou and Georgious [10] derived the analytical solutions of Newton Poiseuille flow in a circular or annular channel and analyzed the effects of opening angle, the radii ratio and the slip number on velocity curves and volume flow. Lee et al. [11] studied the effects of surface roughness on turbulent Couette-Poiseuille flow characteristics and showed that surface roughness has a significant inhibitory effect on Couette-Poiseuille flow on rough walls. Hoyas et al. [12] conducted a numerical simulation of Poiseuille flow and studied the relationship between pressure intensity and Reynolds number. However, the mathematical models established by these studies do not consider the interaction between the fluid flow in gaps and the deformation of the structure.

Other scholars have proposed computational fluid dynamics (CFD) simulation and experimental methods to study the flow characteristics of an annular channel under multiple parameters [13,14]. Qian and Liao [15,16] established a nonisothermal fluid–structure interaction (FSI) mathematical model of piston/cylinder eccentricity and tilt. They concluded that the piston tilt has little effect on leakage, while eccentricity and plunger diameter have been shown to have a great influence on leakage. Nie et al. [17] built a parameterized elastohydrodynamic lubrication model for the piston/cylinder friction pair of seawater hydraulic axial piston pumps and discussed the deformation of piston bushing, bearing mechanism, and energy loss characteristics of the water film under different working conditions. Zhao et al. [18] considered the leakage characteristics of a piston and swiveling cylinder pair of a high water-based hydraulic motor in a one-way FSI interaction. In a deep-sea environment, Li and Wu [19] considered the structural deformation of the clearance fit, the change in a medium viscosity, and the influence of eccentricity and deduced the leakage formula at the clearance fit and conducted verification through a simulation. These scholars have performed a series of studies on annular channel flow. Most scholars focus on the laminar flow in the small-scale low pressure annular gap, and there are few studies on the turbulent flow in the large-scale high-pressure annular channel.

Currently, the existing leakage calculation model of the annular channel is too simplified, resulting in a significant deviation in leakage calculation. Experimental tests have indicated that the leakage rates of turbulent and laminar flow are 1.3 times and 2.5 times that of concentric flow, respectively [20]. Using the parameters in Table 1, leakage of different calculation models at a differential pressure of 10–40 MPa can be calculated, as shown in Figure 1. The results indicate that the leakage ranges of the laminar flow model and turbulent flow model are 128.3–320.8 L/min and 78.2–101.7 L/min, respectively with the deviation of output flow caused by eccentricity reaching 192.5 L/min and 78.2 L/min respectively. The results indicate that eccentricity has a signification influence on the leakage of the laminar and turbulent flow. In addition, studies on the effect of structural deformation and water compressibility of large-scale annular gaps on leakage are rarely mentioned, affecting the accuracy of the leakage model.

To improve the calculation accuracy of the annular channel leakage model, the gap flow equation in laminar and turbulent flow states is established based on the eccentric annular channel. The deformation equation of the annular channel is established based on the pressure gradient inside the plunger pair and the force of the plunger pair. Combined with structural deformation and compressibility of the water medium equations, the leakage equation of the annular channel under fluid-structure interaction is established. Finally, the leakage model was verified by the seal experiment. The research results of this work not only provide a theoretical basis for the accurate design of high-pressure large-scale plunger pumps and have reference significance for reciprocating seal design in other fields.

**Table 1.** Basic parameters of the plunger pump.

| Performance Parameter | Value |
| --- | --- |
| Working pressure [$p$/MPa] | 40 |
| Diameter of plunger [$d$/m] | 0.085 |
| Seal length [$l$/m] | 0.13 |
| Gap [$h_0$/m] | $5 \times 10^{-5}$ |

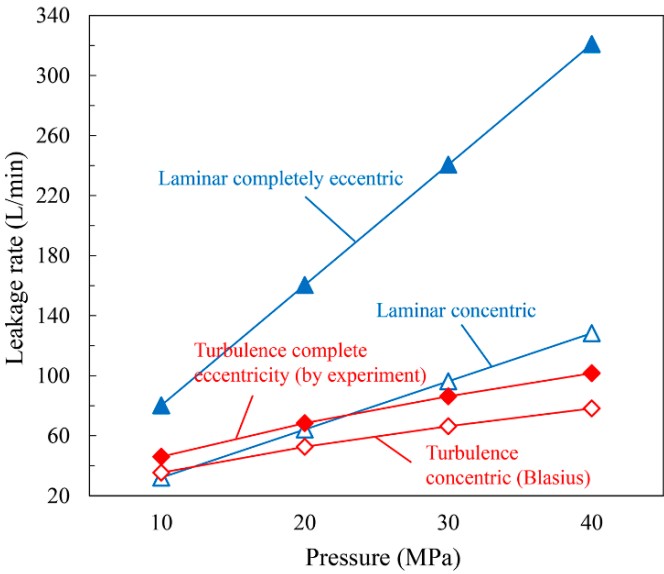

**Figure 1.** Leakage deviation of various calculation models under various pressure differences.

## 2. Working Principle and Parameters of Plunger Pumps

In this work, a five-plunger pump with a flow rate of 1600 L/min and a pressure of 40 MPa is taken as the research object. The structure of the plunger pump is shown in Figure 2, including the crankshaft, connecting rod, slider, plunger, plunger sleeve, cylinder liner, and locking device. When a plunger pump is working, the rotation of the crankshaft drives one end of the connecting rod to make a circular motion. The other end of the connecting rod drives the slider and the plunger to make a linear reciprocating motion in the cylinder liner. During the reciprocating movement of the plunger, the volume of the closed cavity formed by the high-pressure cylinder liner, the plunger sleeve, and the plunger changes, altering the liquid suction and discharge of the plunger pump. In addition, the high-pressure seal of the plunger pump is in the cylinder liner, which seals the high-pressure liquid through the small gap between the plunger sleeve and the plunger.

In engineering, most friction pairs of high-pressure and high-flow plunger pumps adopt hard/hard pairing methods [21,22]. Ceramics have the advantage of low density, low thermal conductivity, and high elastic modulus, while stainless steel shows high corrosion resistance and high strength. Accordingly, the friction pair materials in this study use stainless steel and ceramics. In practice, the plunger pair will undergo elastic deformation

under the action of high-pressure fluid and thermal deformation under heat accumulation. A certain gap for the plunger pair under the dual effects of temperature and pressure field is necessary to ensure that the friction pair does not become stuck during normal operation [23]. When the machining accuracy and assembly requirements of the plunger pump are combined, the appropriate gap and sealing length of the plunger pair is selected for simulation and experiment, as shown in Table 1.

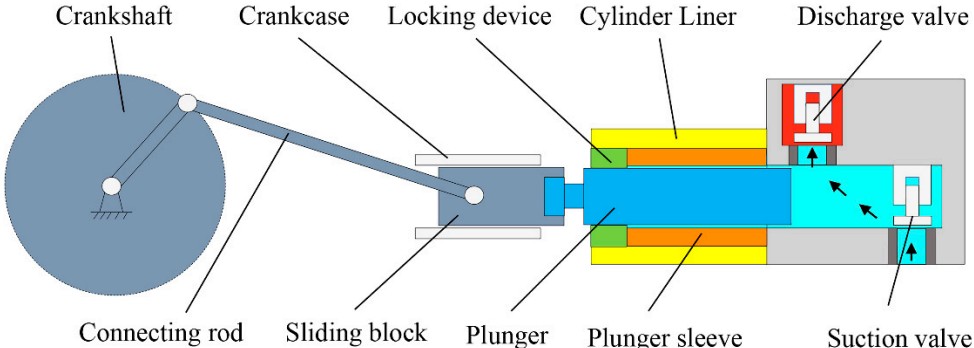

**Figure 2.** Schematic of large-scale high-pressure plunger pump structure.

### 3. Flow Characteristics of the Annular Channel in Complex Operating Conditions

The leakage of the annular channel of the plunger pair is an essential factor affecting the volumetric efficiency of the high-pressure and large-flow plunger pump. As the axial pressure gradient of the high-pressure fluid in the annular channel, the plunger and the plunger sleeve form a funnel-like shape, with a large inlet gap and a small outlet gap. In addition, the plunger and plunger sleeve are generally eccentric because the plunger is subjected to off-load and machining errors in practice. Therefore, based on the eccentric annular channel model, the leakage equation is derived in this chapter. Then, based on the theory of elasticity and the pressure gradient inside the annular gap, the deformation equations of the plunger and plunger sleeve are established. Considering the structural deformation and compressibility of the water medium, the leakage equation of the annular channel under fluid-structure interaction is derived.

*3.1. Annular Channel Flow Model*

In actual work, an eccentricity is likely between the plunger and the plunger sleeve, as shown in Figure 3.

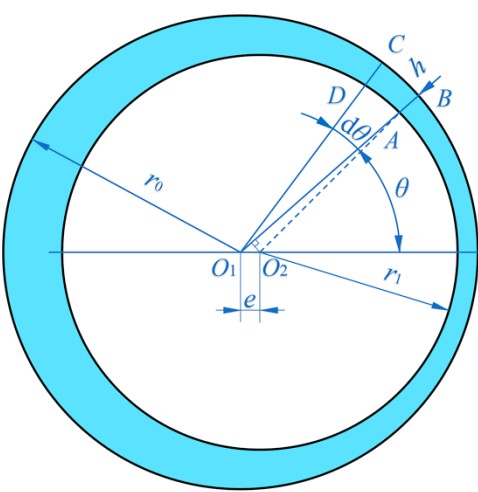

**Figure 3.** Schematic of eccentric annular channel.

To study the flow between the annular channel, the formula of friction resistance coefficient is generally adopted, as follows [5]:

$$\lambda = \frac{p_{\text{in}} - p_0}{l} \frac{2d_1}{\rho v_{\text{a}}^2}. \tag{1}$$

The Reynolds number is generally used to judge the flow state of the fluid using the following equation:

$$\begin{cases} \text{Re} = \frac{\rho v_{\text{a}} d_{\text{h}}}{\mu}, \\ v_1 = \frac{\mu}{\rho}. \end{cases} \tag{2}$$

For laminar flow conditions, the friction resistance coefficient is derived by the Navier-Stokes equation and can be expressed as:

$$\lambda = \frac{64}{\text{Re}}. \tag{3}$$

According to the literature [5,15], the leakage of laminar flow annular channel under an eccentric working condition can be expressed as:

$$q_{\text{c}} = \frac{\pi r_0 h_0^{3}}{6\mu l}(p_{\text{in}} - p_0)\left[1 + 1.5\varepsilon^2\right]. \tag{4}$$

Blasius [24,25] proposed a frictional resistance coefficient of a smooth tube that can be used in the range of Re < 100,000, and can be written as:

$$\lambda = \frac{0.3164}{\sqrt[4]{\text{Re}}}. \tag{5}$$

Combine Equations (1), (2) and (5), the velocity $v_{\text{a}}$ can be written as:

$$v_{\text{a}} = 4.7 h^{\frac{5}{7}}\left(\frac{p_{\text{in}} - p_0}{l\rho}\right)^{\frac{4}{7}}\left(\frac{1}{v_1}\right)^{\frac{1}{7}}. \tag{6}$$

Since the eccentricity $e$ of the plunger pair and the actual gap $h$ are small quantities and vary with the angle $\theta$. Therefore, the following relational expression can be obtained:

$$h = AB = OB - OA \approx r_0 - (r_1 + e\cos\theta) = \delta - e\cos\theta. \tag{7}$$

Taking the section of the angle at $\theta$ as $\mathrm{d}\theta$, and the length of the corresponding small arc CB as $\mathrm{d}y$, CB = $\mathrm{d}y = r_0\mathrm{d}\theta$ can be obtained. Because $\mathrm{d}\theta$ and CB are small, the micro-element gap ABCD can be considered a parallel wall gap with a gap of $h$. By substituting $\mathrm{d}y = r_0\mathrm{d}\theta$ into Equation (7), the leakage flow rate of the eccentric annular channel can be written as follows:

$$q_{\text{t}} = \int_A v_a \mathrm{d}A = \int v_a h\mathrm{d}y = 4.7\left(\frac{p_{\text{in}} - p_0}{l\rho}\right)^{\frac{4}{7}}(v_1)^{-\frac{1}{7}}\int_0^{2\pi}(h)^{\frac{12}{7}}r_0\mathrm{d}\theta. \tag{8}$$

Because the difference between the inner diameter of the plunger sleeve $2r_0$ and the outer diameter of the plunger $2r_1$ is minimal, the radius gap $\delta$ formed by the plunger sleeve and the plunger is also small. Since $\varepsilon = e/h$, and Equation (7) is substituted into Equation (8), the flowrate $q$ of an eccentric annular channel between two parallel walls can also be written as follows:

$$q_{\text{t}} = 4.7 r_0 \delta^{\frac{12}{7}}\left(\frac{p_{\text{in}} - p_0}{l\rho}\right)^{\frac{4}{7}}(v_1)^{-\frac{1}{7}}\int_0^{2\pi}(1 - \varepsilon\cos\theta)^{\frac{12}{7}}\mathrm{d}\theta. \tag{9}$$

Because Equation (9) belongs to the transcendental integral, it is difficult to obtain an analytic solution. Therefore, the Gauss-Lobatto numerical integration method is used to calculate Equation (9) [26]. This study uses the Quadl function and the Vpa precision control function to solve produce an approximate solution for Equation (9) based on the MATLAB software [27]. After the approximate solutions of Equation (9) under different eccentricities are found, the polynomial fitting is performed with the least squares method to fit data using MATLAB software. Furthermore, different sums of squared errors (SSE) for equation $B$ can be obtained by setting different fitting dimensions. The SSE of the data after fitting dimension 2 is very small, reaching only 0.000247. Therefore, the fitting function with a fitting dimension of 2 is taken as the fitting result for consideration of the calculation accuracy and the difficulty of the solution. The eccentric annular channel flow rate can be written as:

$$q_{\mathrm{t}} = 4.7 r_0 h^{\frac{12}{7}} \left( \frac{p_{\mathrm{in}} - p_0}{l\rho} \right)^{\frac{4}{7}} (v_1)^{-\frac{1}{7}} \left( 2.029\varepsilon^2 - 0.0585\varepsilon + 6.288 \right). \tag{10}$$

Because the gap between the plunger pair is tiny, $2r_0$ is approximately equal to $d$, while 6.28 is approximately $2\pi$, so the formula for calculating the turbulent eccentric annular channel caused by the pressure difference can be written as:

$$q_{\mathrm{t}} = 4.7\pi d h^{\frac{12}{7}} \left( \frac{p_{\mathrm{in}} - p_0}{l\rho} \right)^{\frac{4}{7}} (v_1)^{-\frac{1}{7}} \left( 0.3229\varepsilon^2 - 0.0093\varepsilon + 1 \right). \tag{11}$$

*3.2. Structural Deformation of the Plunger Pair*

As indicated in Figure 4, this study analyzed the stress and deformation of a thick-walled cylinder and then derives the deformation of the plunger pair. Because the plunger pair is axisymmetric in space, cylindrical coordinates are used for analysis. According to Equations (4) and (11), in the laminar and turbulent flow state, the pressure of the annular channel is proportional to the sealing length. Therefore, the loading boundary conditions on the cylinder can be expressed as follows:

$$\begin{cases} r = r_{\mathrm{a}}: & \sigma_{\mathrm{r}} = q_1 = k_1 z + p_1, \tau_{\mathrm{rz}} = 0, \\ r = r_{\mathrm{b}}: & \sigma_{\mathrm{r}} = q_2 = k_2 z + p_2, \tau_{\mathrm{rz}} = 0, \\ z = 0, \ l: & \sigma_{\mathrm{z}} = p_3, \ \tau_{\mathrm{rz}} = 0. \end{cases} \tag{12}$$

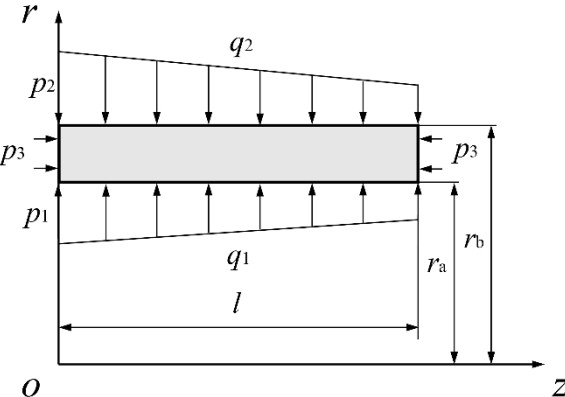

**Figure 4.** Load distribution of the thick-walled cylinder.

The microscopic hexahedron ABCDEFGH is taken from the thick-walled cylinder, as shown in Figure 5. The displacement function $\Phi$ is generally used to solve this spatial axisymmetric problem, so the deformation and stress components can be written as [28]:

$$\begin{cases} \sigma_{\mathrm{r}} = \frac{\partial}{\partial z}\left(\alpha\nabla^2 - \frac{\partial^2}{\partial r^2}\right)\Phi, \\ \sigma_{\theta} = \frac{\partial}{\partial z}\left(\alpha\nabla^2 - \frac{1}{r}\frac{\partial}{\partial r}\right)\Phi, \\ \sigma_{\mathrm{z}} = \frac{\partial}{\partial z}\left[(2-\alpha)\nabla^2 - \frac{\partial^2}{\partial r^2}\right]\Phi, \\ \tau_{\mathrm{rz}} = \frac{\partial}{\partial z}\left[(1-\alpha)\nabla^2 - \frac{\partial^2}{\partial r^2}\right]\Phi. \end{cases} \tag{13}$$

$$\begin{cases} \gamma = -\frac{1}{2G}\frac{\partial^2\Phi}{\partial r\partial z}, \\ w = \frac{1}{2G}\left[2(1-\alpha)\nabla^2 - \frac{\partial^2}{\partial z^2}\right]\Phi. \end{cases} \tag{14}$$

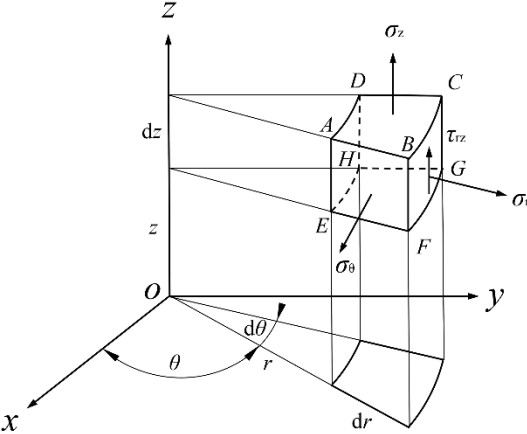

**Figure 5.** Axisymmetric micro-element hexahedron in space.

The Laplace operator $\nabla^2$ and the displacement function $\Phi$ are required to satisfy the biharmonic equation, which can be written as:

$$\begin{cases} \nabla^2 = \frac{\partial^2}{\partial r^2} + \frac{1}{r}\frac{\partial}{\partial r} + \frac{\partial^2}{\partial z^2}, \\ \nabla^2\nabla\Phi = 0. \end{cases} \tag{15}$$

Because the displacement function $\Phi$ must satisfy both the boundary condition Equation (12) and the biharmonic Equation (15), the displacement function can be set as:

$$\Phi = \left[c_1 z^4 + c_2 r^4 + c_3 z^3 + c_4 z^2 r^2 + c_5 z^2 \ln r + c_6 z r^2 + c_7 r^2 \ln r + c_8 z \ln r\right]. \tag{16}$$

The displacement function is obtained through the derivation of (Appendix A), the deformation components $\gamma$ and $w$ can be obtained as follows:

$$\begin{cases} \gamma = -\frac{1}{2G}\left[\begin{array}{l} \frac{\left(k_1 r_{\mathrm{a}}^2 - k_2 r_{\mathrm{b}}^2\right)(\alpha-1)}{(r_{\mathrm{a}}^2 - r_{\mathrm{b}}^2)(\alpha+1)} \times rz - \frac{(k_1-k_2)r_{\mathrm{a}}^2 r_{\mathrm{b}}^2}{(r_{\mathrm{a}}^2 - r_{\mathrm{b}}^2)} \times \left(\frac{z}{r}\right) \\ -\frac{(1-\alpha)\left(p_1 r_{\mathrm{a}}^2 - p_2 r_{\mathrm{b}}^2\right) - \alpha p_3 r_{\mathrm{a}}^2 + \alpha p_3 r_{\mathrm{b}}^2}{(r_{\mathrm{a}}^2 - r_{\mathrm{b}}^2)(\alpha+1)} \times r - \frac{(p_1-p_2)r_{\mathrm{a}}^2 r_{\mathrm{b}}^2}{r_{\mathrm{a}}^2 - r_{\mathrm{b}}^2} \times \frac{1}{r} \end{array}\right], \\ w = -\frac{1}{2G}\left[\begin{array}{l} \frac{\left(k_1 r_{\mathrm{a}}^2 - k_2 r_{\mathrm{b}}^2\right)\alpha}{(r_{\mathrm{a}}^2 - r_{\mathrm{b}}^2)(\alpha+1)} \times z^2 + \frac{\left(k_1 r_{\mathrm{a}}^2 - k_2 r_{\mathrm{b}}^2\right)(1-\alpha)}{2(r_{\mathrm{a}}^2 - r_{\mathrm{b}}^2)(\alpha+1)} \times r^2 \\ +\frac{2\alpha\left(k_1 r_{\mathrm{a}}^2 - k_2 r_{\mathrm{b}}^2\right) - (r_{\mathrm{a}}^2 - r_{\mathrm{b}}^2)p_3}{(r_{\mathrm{a}}^2 - r_{\mathrm{b}}^2)(\alpha+1)} \times z + \frac{(k_1-k_2)r_{\mathrm{a}}^2 r_{\mathrm{b}}^2}{(r_{\mathrm{a}}^2 - r_{\mathrm{b}}^2)} \times (\ln r + 2\alpha) \end{array}\right]. \end{cases} \tag{17}$$

### 3.3. Structural Deformation in the Annular Channel

For the plunger, the deformation caused by the fluid pressure of the annular channel and the end face can be obtained from Equation (17), where $E = E_{\mathrm{p}}$, $k_1 = r_{\mathrm{a}} = p_1 = 0$, $k_2 = -p_{\mathrm{in}}/l$, $p_2 = p_{\mathrm{in}}$, $p_3 = p_{\mathrm{in}}$, $r_{\mathrm{b}} = r_1$, $\alpha = \alpha_{\mathrm{p}}$. Because the radial deformation beyond the clearance fit position of the plunger pair has no influence on leakage, only the radial

deformation of the plunger at the fit clearance needs to be analyzed, and its deformation can be expressed as:

$$L_{\mathrm{p}} = -\frac{\left[1 - 2\alpha_{\mathrm{p}} - (1 - \alpha_{\mathrm{p}})\frac{z}{l}\right]r_1 p_{\mathrm{in}}}{E_{\mathrm{p}}}. \tag{18}$$

For the plunger sleeve, the pressure mainly comes from the annular channel, and its deformation can be obtained from Equation (17). Because the plunger sleeve is fixed on the cylinder liner, the force balance outside of the plunger sleeve will be subjected to the reaction force of the cylinder, and its value is the same as the annular channel pressure. Consequently, $E = E_{\mathrm{s}}$, $k_1 = -p_{\mathrm{in}}/l$, $r_a = r_0$, $p_1 = p_{\mathrm{in}}$, $k_2 = -p_{\mathrm{in}}/l$, $p_2 = p_{\mathrm{in}}$, $p_3 = 0$, $r_b = r_{\mathrm{s}}$, $\alpha = \alpha_{\mathrm{s}}$. Therefore, the deformation of the inner diameter of the plunger sleeve can be expressed as:

$$L_{\mathrm{s}} = \frac{(1 - \alpha_{\mathrm{s}})(r_{\mathrm{s}} - r_0)p_{\mathrm{in}}\left(1 - \frac{z}{l}\right)}{2E_{\mathrm{s}}}. \tag{19}$$

Figure 6 shows the deformation of the plunger and plunger sleeve, such that the fluid flows along the Z direction in the figure. For the plunger, when $z = 0$, the radial deformation of the plunger is negative (along the positive X direction). With increases in $z$, the diameter of the plunger continues to increase, until $z = (1 - 2\alpha_{\mathrm{p}})l/(1 - \alpha_{\mathrm{p}})$, and the plunger deformation is 0. When $z > (1 - 2\alpha_{\mathrm{p}})l/(1 - \alpha_{\mathrm{p}})$, the deformation is positive. For the plunger sleeve, the deformation of the inner hole remains positive and decreases with the increase in $z$. It can be seen from Equations (18) and (19) that the size change rule of the plunger pair along the $z$-axis is linear.

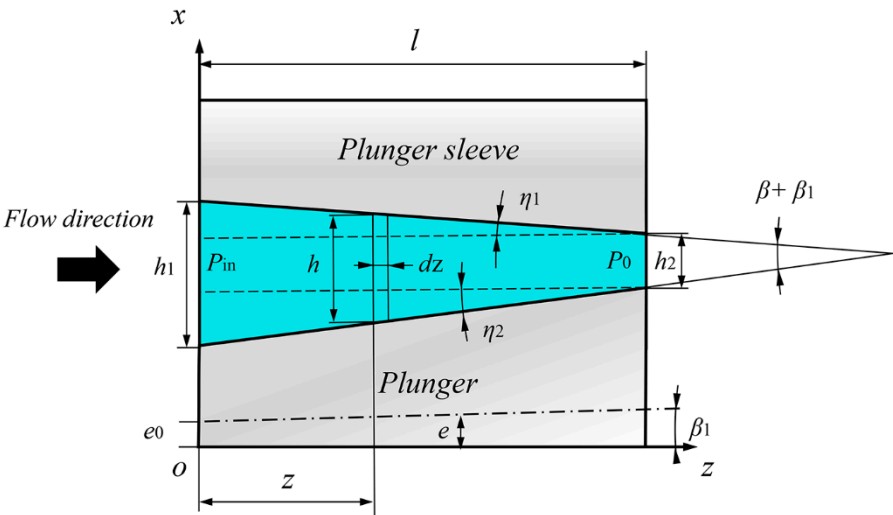

**Figure 6.** Deformation of the annular channel of the plunger pair.

The annular channel deformation of the plunger pair caused by pressure can be expressed as:

$$L_{\mathrm{t}} = \frac{\left[1 - 2\alpha_{\mathrm{p}} - (1 - \alpha_{\mathrm{p}})\frac{z}{l}\right]r_1 p_{\mathrm{in}}}{E_{\mathrm{p}}} + \frac{(1 - \alpha_{\mathrm{s}})(r_{\mathrm{s}} - r_0)p_{\mathrm{in}}\left(1 - \frac{z}{l}\right)}{2E_{\mathrm{s}}}. \tag{20}$$

Due to the small differences between the inner diameter of the plunger sleeve $2r_0$ and the outer diameter of the plunger $2r_1$, the annular channel $h$ of the plunger pair can also be expressed as:

$$h = h_0 + \left\{\frac{(1 - 2\alpha_{\mathrm{p}})}{E_{\mathrm{p}}}r_1 + \frac{(1 - \alpha_{\mathrm{s}})}{E_{\mathrm{s}}}(r_{\mathrm{s}} - r_1)\right\}p_{\mathrm{in}} - \left\{\frac{(1 - \alpha_{\mathrm{p}})}{E_{\mathrm{p}}l}r_1 + \frac{(1 - \alpha_{\mathrm{s}})}{E_{\mathrm{s}}l}(r_{\mathrm{s}} - r_1)\right\}p_{\mathrm{in}}z. \tag{21}$$

According to Equation (21), the annular channel value of the plunger pair decreases linearly along the axial direction. Hence, the height of the annular channel between the inlet and outlet can be written as:

$$\begin{cases} h_1 = h_0 + \left\{ \frac{(1-2\alpha_p)}{E_p}r_1 + \frac{(1-\alpha_s)}{E_s}(r_s - r_1) \right\} p_{in}, \\ h_2 = h_0 - \left( \frac{\alpha_p}{E_p} \right)r_1 p_{in}. \end{cases} \tag{22}$$

The unit of the fitting length is the millimeter, but the plunger pair gap is micrometers. Therefore, the deformation of the plunger pair is far less than the fitting length, and the cone angle of the annular channel is also a small amount. The following relation can be obtained:

$$\begin{cases} \tan\beta = \tan(\eta_1 + \eta_2) \approx \tan\eta_1 + \tan\eta_2, \\ h = h_1 - z\tan(\beta + \beta_1) = h_1 - z(\tan\beta + \tan\beta_1), \\ e = e_0 - z\tan\beta_1. \end{cases} \tag{23}$$

### 3.4. Compressible Water Medium Equation

Generally, water is considered an incompressible medium, but the volume loss due to the compressibility of water under high-pressure conditions cannot be ignored. Considering the bulk modulus of water and the continuity equation of flow, the following equation can be expressed:

$$k_w = -dp\frac{V}{dV} = dp\frac{\rho}{d\rho} = \rho\frac{p}{\rho - \rho_0}. \tag{24}$$

### 3.5. Leakage Equation of Plunger Pair under Complex Working Conditions

The volume flow cannot be used to measure leakage after taking the compressibility of water into account. Considering the structural deformation and eccentricity of the plunger pair and compressibility of the water, the mass flow rate of leakage from the annular channel of the plunger pair under the laminar flow can be calculated with the following:

$$q_{cw} = q\rho = \frac{k_w\rho_0}{k_w - p}\frac{\pi r_0(h_1 - z\tan\beta - z\tan\beta_1)^3}{6\mu l dz}dp\left[1 + 1.5\left[\frac{e_0 - \tan\beta_1}{h_1 - z(\tan\beta + \tan\beta_1)}\right]^2\right]. \tag{25}$$

The derivation of the Equations (see Appendix B), because $dz = dh/(\tan\beta)$, the mass flow rate of laminar leakage can be obtained as follows:

$$\begin{aligned} q_{cw} &= \frac{-\frac{\pi r_0(k_w\rho_0)}{6\mu}[\ln(k_w - p_0) - \ln(k_w - p_{in})]\left[1 + 1.5\left[\frac{2e_0}{(h_1 + h_2)}\right]^2\right]}{\int_{h_1}^{h_2}\frac{-ldz}{(h_1 - h_2)h^3}} \\ &= \frac{\pi r_0 k_w\rho_0 h_1{}^2 h_2{}^2}{3\mu l(h_1 + h_2)}[\ln(k_w - p_0) - \ln(k_w - p_{in})]\left[1 + 1.5\left[\frac{2e_0}{(h_1 + h_2)}\right]^2\right]. \end{aligned} \tag{26}$$

Similar to laminar flow, the mass flow of an annular channel in a turbulent flow state can be expressed as:

$$q_{tw} = q\rho = \left\{ \begin{aligned} &4.7\pi d[h_1 - z(\tan\beta + \tan\beta_1)]^{\frac{12}{7}}\left(\frac{dp}{dz}\right)^{\frac{4}{7}}\left(\frac{k_w\rho_0}{k_w - p}\right)^{\frac{3}{7}}\left(\frac{k_w\rho_0}{(k_w - p)\mu}\right)^{\frac{1}{7}} \\ &\times \left[0.3229\left[\frac{e_0 - \tan\beta_1}{h_1 - z(\tan\beta + \tan\beta_1)}\right]^2 - 0.0093\left[\frac{e_0 - \tan\beta_1}{h_1 - z(\tan\beta + \tan\beta_1)}\right] + 1\right] \end{aligned} \right\}. \tag{27}$$

Similar to laminar flow, the average eccentricity of the inlet and outlet is used to replace the eccentricity, and the leakage of the turbulent annular channel can be obtained as follows:

$$q_{tw} = 9.4\pi r_1\left(\frac{1}{\mu}\right)^{\frac{1}{7}}\left\{ \frac{2h_1{}^2 h_2{}^2 k_w\rho_0}{(h_1 + h_2)l}[\ln(k_w) - \ln(k_w - p_{in})] \right\}^{\frac{4}{7}} \times \left[0.3229\left(\frac{2e_0}{(h_1 + h_2)}\right)^2 - 0.0093\left(\frac{2e_0}{(h_1 + h_2)}\right) + 1\right]. \tag{28}$$

## 4. Test System

A gap seal test bench is built to study the leakage of the plunger pair under real working conditions [29–32].

### 4.1. Working Principle of Sealing Test Bench

Figure 7 shows the schematic of the gap seal test bench system, which includes three main parts: a flow test device, a water hydraulic system, and a hydraulic system. The flow test device is an important part of the sealing test bench and is divided into two areas. The first area is the plunger cavity, composed of a cylinder liner and a plunger pair, which is provided with a high-pressure environment by the pump station. The second area is the piston rod coaxial, with the plunger, which drives the plunger pair to reciprocate with the help of hydraulic pressure. In the second part, the water hydraulic system provides a high-pressure fluid environment for the sealing test with a three-plunger pump. In the third part, the hydraulic system is composed of a servo cylinder, a plunger pump, a servo valve, and other pieces, which together can accurately control the reciprocating frequency and action amplitude of the plunger.

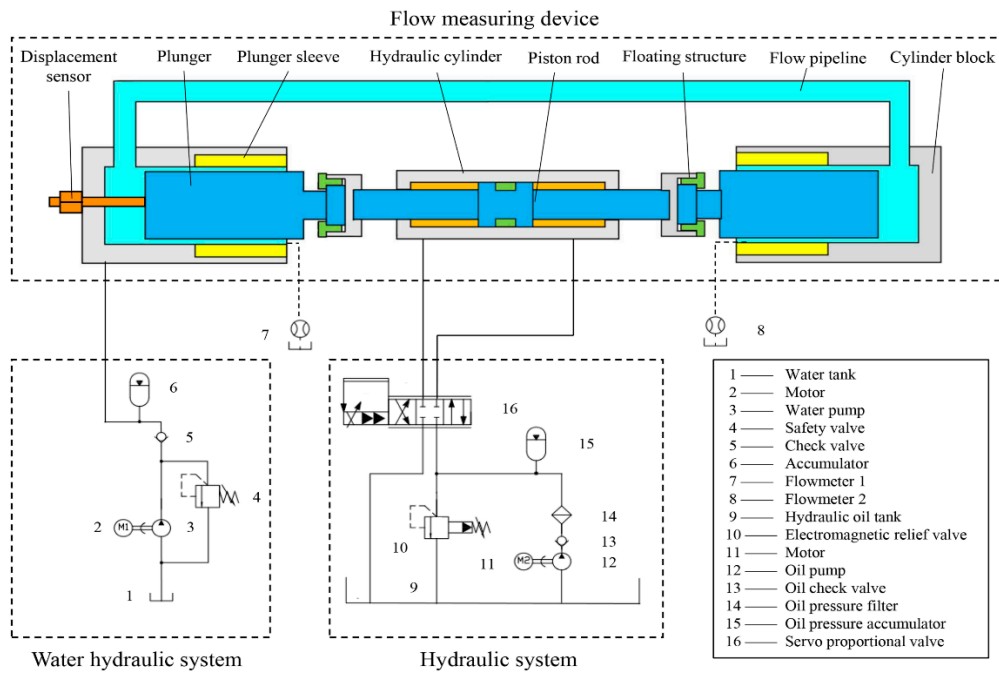

**Figure 7.** Schematic of clearance seal test bench system.

Figure 8 represents the picture of the sealing test bench. The hydraulic pressure generated by the liquid on the plunger can be offset by a symmetrically arranged structure because the two plunger cavities in the flow test device are connected through the flow pipeline. Therefore, this test bench can not only change the plunger position by hydraulic cylinder to conduct leakage tests with different seal lengths but also greatly reduce the driving power and experimental cost. The basic parameters of the sensor used in the test bed are shown in Table 2.

**Table 2.** Basic parameters of the sealing test bench.

| Parameter | Value |
| --- | --- |
| Range of flowmeter (L/min) | 6.8–68 |
| Accuracy of flowmeter | ±0.2%FS |
| Range of temperature sensor (°C) | −50–100 |
| Accuracy of temperature sensor | ±0.5%FS |

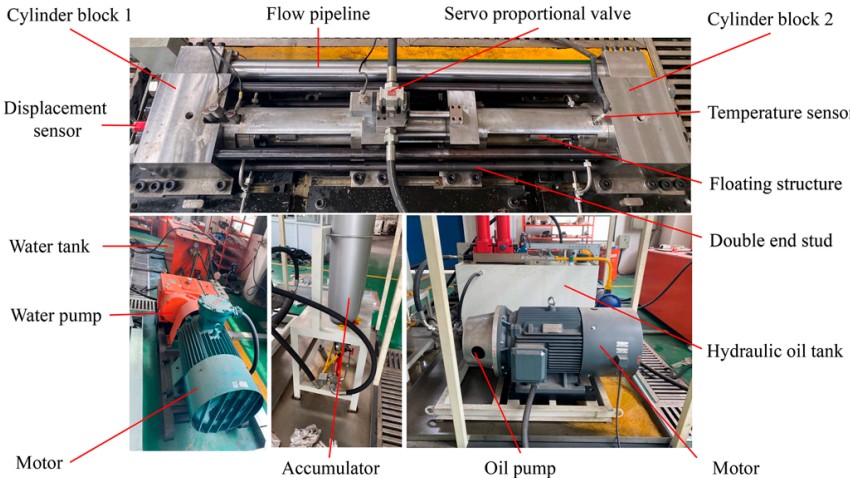

**Figure 8.** Picture of a high-speed large-scale reciprocating gap-seal test bench.

*4.2. Eccentricity and Leakage Test*

Figure 9 shows a schematic diagram of the eccentricity test of the plunger pair. The plunger is first fixed by tightening the floating structure, and then the locking device is removed. The gaps at points *A* and *B* are measured in three directions using a feeler gauge with an accuracy of 0.001–0.1 mm, and then the locking device was reinstalled.

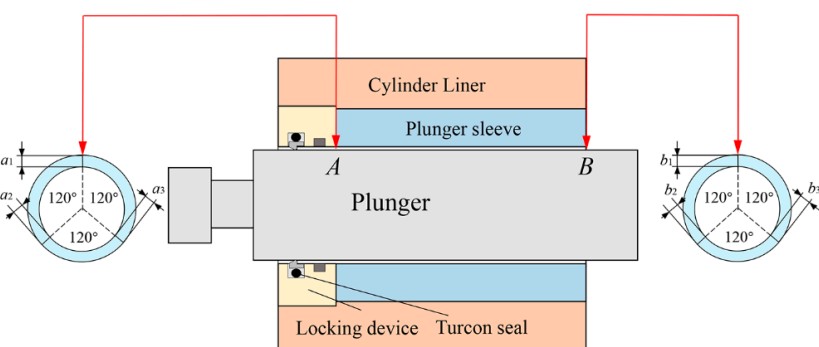

**Figure 9.** Schematic diagram of eccentricity test.

To eliminate the error caused by temperature, the temperature sensor is used to monitor the plunger temperature. The specific steps of the sealing test bench are as follows. First, the plunger is driven by the hydraulic cylinder to the end position, and we test the eccentricities of both ends of the plunger pair. Then the water hydraulic system is turned on to supply liquid to the test device, and the required pressure is loaded to ensure that there is no leakage at any part, and the data from the flowmeter are recorded. The basic parameters of the plunger pair used in the experiment are shown in Table 3.

**Table 3.** The parameters of the plunger pair.

| Parameter | Type | Valve |
|---|---|---|
| Plunger | $R_a$ | 0.03 |
| | $d$ | $84.901 \times 10^{-3}$ |
| Plunger sleeve | $R_a$ | 0.18 |
| | $2r_0$ | $85.003 \times 10^{-3}$ |
| Position A | $a_1$ (m) | $4.7 \times 10^{-8}$ |
| | $a_2$ (m) | $6.6 \times 10^{-8}$ |
| | $a_3$ (m) | $3.7 \times 10^{-8}$ |
| Position B | $b_1$ (m) | $4.3 \times 10^{-8}$ |
| | $b_2$ (m) | $6.6 \times 10^{-8}$ |
| | $b_3$ (m) | $4.1 \times 10^{-8}$ |

At least three repeated tests are performed on the leakage of the plunger pair with a gap of 0.05 mm and a pressure of 10–40 MPa. The difference between the results of multiple experiments under various pressure differences is slight. This result indicates that the repeatability of the experimental bench is good, and the test results are accurate and reliable. The plunger pair temperature before and after the experiment was 25.7 °C and 25.3 °C, respectively.

## 5. Results and Discussion

### 5.1. Experimental Verification

The material and fluid properties of the plunger pair are set as presented in Table 4.

**Table 4.** Material parameters of the water and the plunger.

| Parameter | Type | Valve |
|---|---|---|
| Water | $\rho_0$ (kg/m$^3$) | 998.2 |
| | $\mu$ (Pa·s) | 0.001 |
| | $k_w$ (GPa) | 2.18 |
| Plunger | $E_p$ (GPa) | 202 |
| | $\alpha_p$ | 0.3 |
| | $E_s$ (GPa) | 300 |
| Plunger sleeve | $\alpha_s$ | 0.29 |
| | $r_s$ (m) | 0.055 |

From the above, the eccentricities of the plunger calculated from Table 3 are 0.34 and 0.3209, respectively, and the inclination angle is 0.0041°. It can be found that the inclination of the plunger pair in the plunger hole is slight. Many scholars [7,15] believe that the influence of plunger tilt on leakage is far less than that of eccentricity, so the effect of inclination on leakage can be ignored. The average eccentricity of the plunger 0.3304 was substituted into the formula of the laminar flow and turbulence, and the leakage data obtained was compared with the experiment, as shown in Figure 10. At 10 MPa pressure, the deviation of the laminar flow model and turbulence model from the experiment is small, 1.05% and 1.58%, respectively. The laminar flow model significantly differs from the experiment at the pressure range of 20–40 MPa. In contrast, the turbulence model has a slight deviation of 0.277%, 4.32%, and 5.59%, respectively at the pressure range of 20–40 MPa. In conclusion, the derived formulas of laminar flow and turbulence have good accuracy and stability. It can accurately predict the leakage of annular gaps in large-scale high-pressure plunger pairs under different eccentricities.

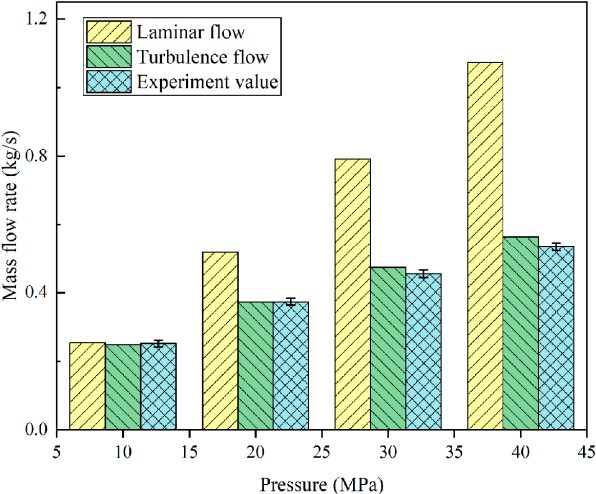

**Figure 10.** Comparison of different leakage models.

### 5.2. Proportion of Factors Affecting the Leakage

Based on different leakage models, this section explores the effects of water compressibility, eccentricity and structural deformation on leakage. When the pressure of the plunger pair is 10 MPa, laminar flow and turbulence are used to calculate the leakage under different eccentricity and gap heights, as shown in Figure 11. From the figure, the leakage of annular channels in both laminar flow and turbulence increases with the increase of eccentricity, and the growth rate of laminar flow is greater than that of turbulence. When the eccentricity is 0–0.29, the leakage of laminar flow is smaller than that of turbulent flow. When the eccentricity is 0.29–1, the leakage of laminar flow is greater than that of turbulent flow. It is worth noting that, compared with the concentric leakage, the total eccentric leakage in laminar flow is 2.48 times of the former. In turbulent flow, it is 1.31 times.

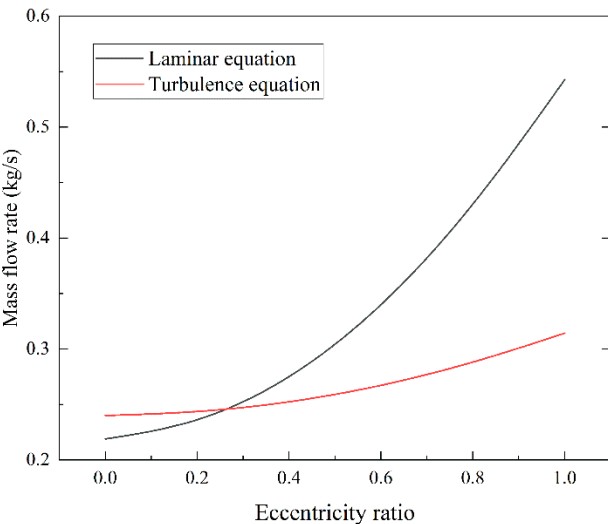

**Figure 11.** Leakage of the laminar and turbulence flow under different eccentricities.

Taking a concentric annular channel with a gap of 0.05 mm as an example, the inlet and outlet heights $h_{1w}$, $h_{2w}$, and leakage of the plunger pair under different pressures and flow modes are calculated, as shown in Table 5. After considering the structural deformation, the inlet gap of the annular channel becomes larger and the outlet gap becomes smaller. The medium compressibility has little effect on the size of the annular gap. In addition, by comparing different models, it can be considered that the influence of structural deformation and compressibility on the laminar flow model is more significant than that on the turbulence model. Under the laminar flow condition, the influences of structural deformation and compressibility under 10–40 MPa pressure are 1.7%, 3.23%, 4.61% and 5.84%, respectively. In the turbulent flow state, they are 1.63%, 2.51%, 3.29% and 4.03%, respectively.

**Table 5.** The inlet and outlet heights and leakage of the annular channel.

| Parameter | Cond.1 | Cond.2 | Cond.3 | Cond.4 |
|---|---|---|---|---|
| $p_{in}$ (MPa) | 10 | 20 | 30 | 40 |
| $h_{1w}$ (m) | $5.114 \times 10^{-5}$ | $5.227 \times 10^{-5}$ | $5.341 \times 10^{-5}$ | $5.455 \times 10^{-5}$ |
| $h_{2w}$ (m) | $4.937 \times 10^{-5}$ | $4.874 \times 10^{-5}$ | $4.811 \times 10^{-5}$ | $4.747 \times 10^{-5}$ |
| $q_c$ (kg/s) | 0.2139 | 0.4277 | 0.6416 | 0.8555 |
| $q_{cw}$ (kg/s) | 0.2176 | 0.442 | 0.6726 | 0.9086 |
| $q_t$ (kg/s) | 0.2353 | 0.3497 | 0.4409 | 0.5196 |
| $q_{tw}$ (kg/s) | 0.2392 | 0.3587 | 0.4559 | 0.5414 |

This section explores the influence of compressibility of high-pressure water medium and structural deformation on leakage based on different leakage models, as shown in

Figure 12. It can be concluded that the trend of calculated leakage under the laminar flow and turbulent flow is the same. After considering the compressibility of the water medium and structural deformation, the leakage of the annular channel increases. In general, the influence of structural deformation on leakage is more significant than the compressibility of the water medium, and the impact is different in different flow states. It is reported that the influence of the structural deformation of the plunger pair on the leakage rate is 3.9% and 4.92% under the laminar flow regime of 30 MPa and 40 MPa, respectively. The effects of 30 MPa and 40 MPa in a turbulent flow regime are 2.25% and 2.84%, respectively. It can be suggested that the compressibility of water has little influence on the leakage, and the influence of 30 MPa and 40 MPa in laminar flow is 0.69% and 0.92%, respectively. The effects of 30 MPa and 40 MPa in the turbulent flow regimes are 1.04% and 1.19%, respectively.

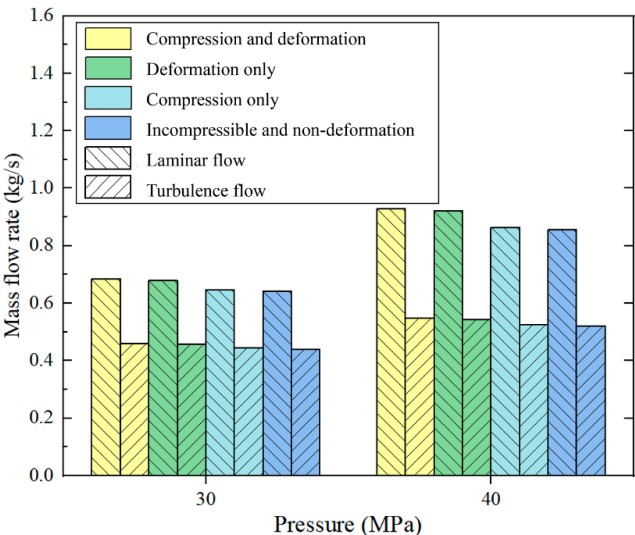

**Figure 12.** Effect of compressibility and structural deformation on leakage.

For a large-flow plunger pump, the volumetric efficiency of the pump needs to reach 88% at 40 MPa. After evaluation, assuming that the volumetric efficiency of the rest of the pump remains unchanged, the leakage of the gap seal needs to be controlled within 5% to ensure the efficiency of the pump. A single plunger leakage cannot exceed 0.607 kg/s at 40 MPa. When the leakage of the plunger pair is 0.607 kg/s, the parameters of the plunger diameter, sealing length, and gap height are shown in Figure 13. It can be concluded that the influencing factors on the leakage are gap, plunger diameter, and seal length in descending order.

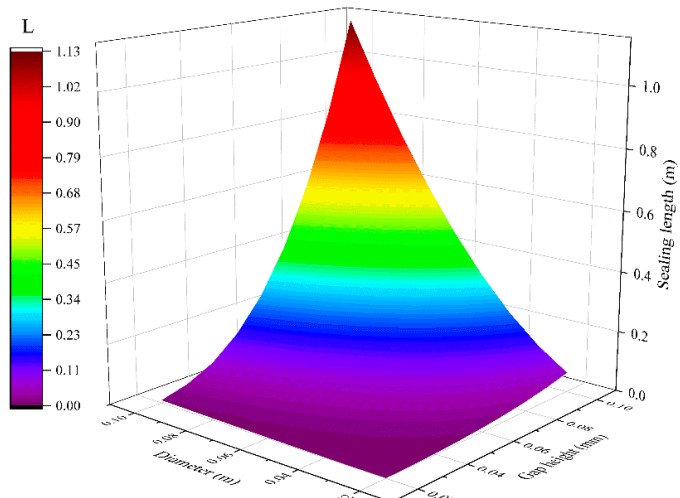

**Figure 13.** Parameter matching of different leakage influencing factors.

### 5.3. Influence of Material Properties on Leakage

The leakage of plunger pairs with different materials is studied using the mathematical model in this study. The selected materials of the plunger pair are commonly used materials in a water medium, such as 17-4PH stainless steel, Si3N4, aluminum bronze (QAL9-4), titanium alloy (TC4), and aluminum alloy (LD5), etc., and its physical properties are shown in Table 6 [33].

**Table 6.** Physical properties of materials.

| Material | $\alpha$ | $E$ (GPa) |
|:---:|:---:|:---:|
| 17-4PH | 0.3 | 202 |
| Si3N4 | 0.29 | 300 |
| QAL9-4 | 0.49 | 116 |
| TC4 | 0.34 | 108 |
| LD5 | 0.33 | 72 |

The leakage of the 17-4PH stainless steel plunger is compared with other plunger sleeves of different materials based on 40 MPa pressure and turbulence-concentric working conditions, as shown in Figure 14. It is found that the leakage of plunger pair from large to small is LD5, TC4, QAL9-4, Si3N4. For plunger sleeves of different materials, the leakage decreases with the increase of Poisson's ratio and elastic modulus. The influence of elastic modulus on the leakage is more significant than Poisson's ratio.

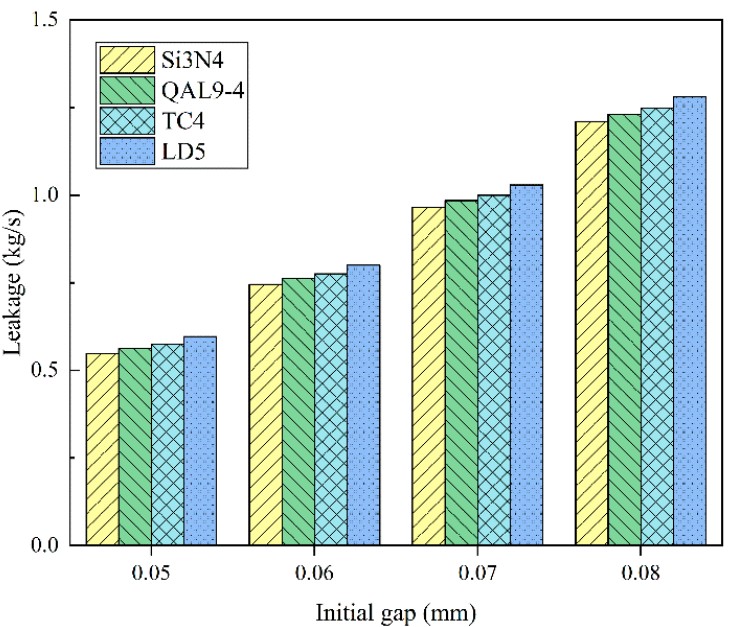

**Figure 14.** Different material plunger sleeves are paired with 17-4PH plungers.

Figure 15 shows the leakage of the Si3N4 plunger sleeve paired with other plungers of different materials. It is found that the leakage of the plunger pair from large to small is 17-4PH, TC4, LD5, and QAL9-4. For plungers of different materials, with the increase of Poisson's ratio of the plunger, the gap height between the inlet and outlet decreases, resulting in the decrease of leakage. With the increase of elastic modulus, the inlet gap $h_1$ decreases, and the outlet gap $h_2$ increases. It can be seen from the formula that the decreasing amplitude of $h_1$ is positively correlated with $1 - 2\alpha_p$, while the increasing amplitude of $h_2$ is positively correlated with $\alpha_p$. Therefore, the influence of elastic modulus on leakage depends on the values of these two values. When $1 - 2\alpha_p > \alpha_p$, the elastic modulus increases, the overall gap height decreases, and the leakage decreases, whereas the leakage increases. In conclusion, increasing Poisson's ratio can reduce the clearance

height of the plunger pair and reduce the leakage, and the relationship between the elastic modulus and the leakage depends on Poisson's ratio.

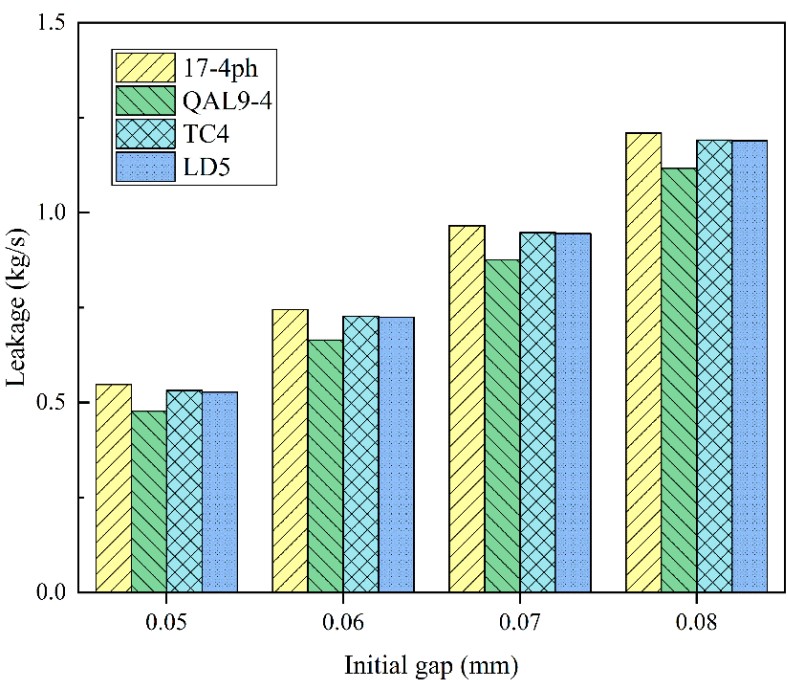

**Figure 15.** Different material plungers are paired with Si3N4 plunger sleeves.

## 6. Conclusions

In this study, the effects of plunger eccentricity, structural deformation, and fluid compressibility on the annular channel flow characteristics are studied by theory and experiment. The main conclusions are as follows:

1.  From the pressure gradient inside the annular gap and elasticity theory, the deformation equation of the annular channel of the plunger pair was derived. The leakage of the annular gap of the plunger pair under laminar and turbulent flow is derived based on the compressibility of water. According to the equation, the annular gap forms a funnel-like shape after deformation, with a large gap height at the inlet and a small gap height at the outlet.

2.  From the eccentricity, structural deformation and compressibility, the leakage equation of the annular gap is derived. According to the equation, the complete eccentricity of laminar flow is 2.48 times that of concentric flow. In turbulent flow, the sensitivity of leakage to eccentricity is low, and it is about 1.31 times concentric when it is completely eccentric. The influence of structural deformation and compressibility under laminar flow on leakage is 4.92% and 0.92%, respectively. In turbulent conditions, they were 2.84% and 1.19%, respectively.

3.  A sealing test bench is built to test the eccentricity and leakage of the plunger under different pressures. Comparing the calculated values with the experimental results, it is found that the maximum deviation is less than 6%. The results indicate that the derived formula of laminar flow and turbulence can accurately predict the leakage of the annular channel, which can provide a theoretical basis for designing large-scale high-pressure reciprocating seals.

4.  The results show that the influencing factors of leakage from large to small are: gap height, plunger diameter, seal length, eccentricity, structural deformation, and water compressibility.

5.  The friction pair of the plunger needs a reasonable material pairing. It is found that the plunger sleeve with a large Poisson's ratio and elastic modulus, and the plunger sleeve with a large Poisson's ratio can effectively reduce the gap height and leakage.

The relationship between elastic modulus and leakage of the plunger depends on Poisson's ratio. In addition, Poisson's ratio has a greater influence on leakage than elastic modulus for plungers of different materials, and vice versa for plunger sleeves.

**Author Contributions:** S.Z. wrote the main manuscript text. Y.L. provided suggestions for improvement of the manuscript. D.W., C.W. and Z.W. helped to set up the experimental apparatus. All authors have read and agreed to the published version of the manuscript.

**Funding:** The authors thank the National Natural Science Foundation of China (No.52075192) and Jiangsu Province competitive Projects (BE2020087) for providing the funding for this project.

**Data Availability Statement:** All data generated or analyzed during this study are included in this article.

**Conflicts of Interest:** The authors have no relevant financial or non-financial interests to disclose.

### Nomenclature

| | | | |
|---|---|---|---|
| $A$ | the area of the annular channel (m$^2$) | $r_1$ | plunger radius (m) |
| $c_1$–$c_8$ | undetermined coefficients | $r_s$ | outer wall radius of plunger sleeve (m) |
| $d$ | diameter of the plunger (m) | Re | Reynolds number |
| $d\theta$ | the angle corresponding to the arc CB (rad) | $R_a$ | surface roughness (μm) |
| $dy$ | the length of the arc CB (m) | $p$ | working pressure (MPa) |
| $d_1$ | the diameter of the pipe | $p_{in}$ | inlet pressure (MPa) |
| $d_h$ | hydraulic diameter (m) | $p_0$ | outlet pressure (MPa) |
| $e$ | eccentric value of plunger (m) | $p_1$ | inner pressure of the cylinder (MPa) |
| $e_0$ | the eccentricity at $z = 0$ | $p_2$ | outer pressure of the cylinder (MPa) |
| $E_p$ | elastic modulus of the plunger (MPa) | $p_3$ | axial pressure of the cylinder (MPa) |
| $E_s$ | elastic modulus of plunger sleeve (MPa) | $v_a$ | average flow velocity (m/s) |
| $G$ | shear modulus (MPa) | $v_1$ | kinematic viscosity (m$^2$/s) |
| $h$ | gap (m) | $z$ | axial coordinate value (m) |
| $h_0$ | the initial gap of the plunger pair (m) | $\alpha$ | Poisson's ratio |
| $h_1$ | the gap at inlet (m) | $\alpha_p$ | Poisson's ratio of the plunger |
| $h_2$ | the gap at the outlet (m) | $\alpha_s$ | Poisson's ratio of the plunger sleeve |
| $h_{1w}$ | the gap at the inlet after considering structural deformation (m) | $\beta$ | the cone angle of the annular channel |
| $h_{2w}$ | the gap at the inlet after considering structural deformation (m) | $\beta_1$ | the inclination angle of the annular channel |
| $k_1$ | the slope of the inner linear load | $\gamma$ | radial deformation component (m) |
| $k_2$ | the slope of the outer linear load | $\delta$ | the average height of the annular channel (m) |
| $k_w$ | the bulk modulus of water (GPa) | $\varepsilon$ | the eccentricity of the plunger |
| $l$ | seal length (m) | $\eta_1$ | the cone angle of the plunger sleeve |
| $L_p$ | the radial deformation of the plunger (m) | $\eta_2$ | the cone angle of the plunger |
| $L_s$ | the radial deformation of the plunger sleeve (m) | $\theta$ | any angle on a circle (rad) |
| $L_t$ | total deformation of annular channels (m) | $\lambda$ | coefficient of frictional resistance |
| $q$ | volume flow rate (m$^3$/s) | $\mu$ | dynamic viscosity (Pa·s) |
| $q_c$ | mass flow rate under laminar flow (kg/s) | $\rho_0$ | the density of water at standard atmospheric pressure (kg/m$^3$) |
| $q_t$ | mass flow rate under turbulence flow (kg/s) | $\rho$ | the density of the water (kg/m$^3$) |
| $q_{cw}$ | laminar flow of structural deformation and water compression (kg/s) | $\sigma_r$ | the radial component of stress (MPa) |
| $q_{tw}$ | turbulence flow of structural deformation and water compression (kg/s) | $\sigma_\theta$ | circumferential stress component (MPa) |
| $q_1$ | inner linear load (N) | $\sigma_z$ | axial stress component (MPa) |
| $q_2$ | outer linear load (N) | $\tau_{rz}$ | tangential shear stress (MPa) |
| $r$ | radial coordinate value (m) | $\Phi$ | displacement function |
| $r_a$ | the inner radius of the cylinder (m) | $w$ | axial deformation component (m) |
| $r_b$ | the outer radius of the cylinder (m) | $\nabla^2$ | three-dimensional Laplace operator |
| $r_0$ | the radius of the plunger sleeve (m) | | |

**Appendix A. Derivation of Equation (17)**

If we substitute Equations (16) and (15) into Equation (12), the stress components can be rewritten as follows:

$$\begin{cases} \sigma_r = \left[24c_1 z\alpha + 6c_3\alpha + 4c_4 z(2\alpha - 1) + 2c_5\frac{1}{r^2}z + 2c_6(2\alpha - 1) + c_8\frac{1}{r^2}\right], \\ \sigma_\theta = \left[24c_1 z\alpha + 6c_3\alpha + 4c_4 z(2\alpha - 1) - 2c_5\frac{1}{r^2}z + 2c_6(2\alpha - 1) + c_8\frac{1}{r^2}\right], \\ \sigma_z = \left[24c_1 z(1-\alpha) + 6c_3(1-\alpha) + 8c_4 z(2-\alpha) + 4c_6(2-\alpha)\right], \\ \tau_{rz} = \left[32c_2 r(1-\alpha) - 4c_4 r\alpha - 2c_5\frac{1}{r}\alpha + 4c_7\frac{1}{r}(1-\alpha)\right]. \end{cases} \quad (A1)$$

If we substitute Equation (A1) into Equation (12) because the coefficients in front of functions of the same order on both sides of the equation should be the same, the following equation can be obtained:

$$\begin{cases} 24c_1\alpha + 4c_4(2\alpha - 1) + 2c_5\frac{1}{r_a^2} = k_1, \\ 24c_1\alpha + 4c_4(2\alpha - 1) + 2c_5\frac{1}{r_b^2} = k_2, \\ 6c_3\alpha + 2c_6(2\alpha - 1) + c_8\frac{1}{r_a^2} = p_1, \\ 6c_3\alpha + 2c_6(2\alpha - 1) + c_8\frac{1}{r_b^2} = p_2, \\ 6c_3(1-\alpha) + 4c_6(2-\alpha) = p_3, \\ 3c_1(1-\alpha) + c_4(2-\alpha) = 0, \\ 8c_2(1-\alpha) - c_4\alpha = 0, \\ 2c_7(1-\alpha) - c_5\alpha = 0. \end{cases} \quad (A2)$$

By solving Equation (A2), the undetermined coefficients $c_1$–$c_8$ can be obtained, as follows:

$$\begin{cases} c_1 = \frac{(2-\alpha)\left(k_1 r_a^2 - k_2 r_b^2\right)}{12(r_a^2 - r_b^2)(\alpha+1)}, c_2 = -\frac{\alpha\left(k_1 r_a^2 - k_2 r_b^2\right)}{32(r_a^2 - r_b^2)(\alpha+1)}, \\ c_3 = \frac{(4-2\alpha)\left(p_1 r_a^2 - p_2 r_b^2\right) + (1-2\alpha)\left(r_a^2 - r_b^2\right)p_3}{6(r_a^2 - r_b^2)(\alpha+1)}, \\ c_4 = \frac{\left(k_1 r_a^2 - k_2 r_b^2\right)(\alpha-1)}{4(r_a^2 - r_b^2)(\alpha+1)}, c_5 = -\frac{(k_1 - k_2)r_a^2 r_b^2}{2(r_a^2 - r_b^2)}, \\ c_6 = -\frac{(1-\alpha)\left(p_1 r_a^2 - p_2 r_b^2\right) - \alpha p_3 r_a^2 + \alpha p_3 r_b^2}{2(r_a^2 - r_b^2)(\alpha+1)}, \\ c_7 = \frac{\alpha r_a^2 r_b^2(k_1 - k_2)}{4(r_a^2 - r_b^2)(\alpha-1)}, c_8 = -\frac{(p_1 - p_2)r_a^2 r_b^2}{r_a^2 - r_b^2}. \end{cases} \quad (A3)$$

**Appendix B. Derivation of Equation (26)**

To facilitate the calculation, the equation can be rewritten as:

$$\frac{dz}{(h_1 - z\tan\beta - z\tan\beta_1)^3\left[1 + 1.5\left[\frac{e_0 - \tan\beta_1}{h_1 - z(\tan\beta + \tan\beta_1)}\right]^2\right]} = \frac{k_w\rho_0}{k_w - p}\frac{\pi r_0}{6\mu l q_{cw}}dp. \quad (A4)$$

By integrating the two sides of the formula, the following formula can be obtained:

$$\int_0^l \frac{dz}{(h_1 - z\tan\beta - z\tan\beta_1)^3\left[1 + 1.5\left[\frac{e_0 - \tan\beta_1}{h_1 - z(\tan\beta + \tan\beta_1)}\right]^2\right]} = \frac{\pi r_0(k_w\rho_0)}{6\mu l q_{cw}}\int_{p_{in}}^{p_0}(k_w - p)^{-1}dp. \quad (A5)$$

Because the definite integral in Equation (A5) is challenging to obtain an analytical solution, and because the annular gap is minimal, the influence of inclination on the leakage is much less than that of eccentricity [15]. As the gap change caused by structural

deformation is small when $z$ goes from 0 to $l$, when the eccentricity is approximate to the average eccentricity at its inlet and outlet [19], which can be written as follows:

$$\frac{e_0}{h_1 - z \tan \beta} = \frac{2e_0}{(h_1 + h_2)}. \tag{A6}$$

The inclination angle is treated as $\beta_1 = 0$, therefore, Equation (A5) can be written as:

$$\int_0^l \frac{dz}{(h_1 - z \tan \beta)^3 \left[1 + 1.5\left[\frac{2e_0}{(h_1 + h_2)}\right]^2\right]} = -\frac{\pi r_0 (k_w \rho_0)}{6\mu l q_{cw}}[\ln(k_w - p_0) - \ln(k_w - p_{in})]. \tag{A7}$$

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
