# Peer review of "Theoretical and Experimental Flow Characteristics of a Large-Scale Annular Channel in Terms of Deformation Gradient, Eccentricity, and Water Compressibility"

_lubricants, doi:10.3390/lubricants11030134_

Round 1

Reviewer 1 Report

the manuscript gives a brief account on Flow Characteristics of an annular channeland it is of current research interest. however, the manuscript has many flaws before considering it for publication. 
  1. The abstract has no proper focus on the core findings of the research. Hence, the abstract should be modified with the outcomes of the investigation.   
  2. The novelty statement should be included and properly justified from the research gaps.  
  3. The justification for considering a low rate of 1600 L/min and a pressure of 40 MPa is not properly elucidated in the manuscript. It is mentioned that parameters are considered from the literature. It should be included “what made authors to chose particular operating parameters?” What happens to the model if the operating parameters are not in the range of assumptions?  
  4. The methodology should be given in a flowchart. Also, the derivations can be given in the annexure (as it consumes more length as well as they already exist in the literature). Only the important equations can be seen in the manuscript flow.  
  5. The cause-and-effect relations of Figures 10, 11 and 12 can be included in the manuscript. Also, the error bars can be included for the outcomes of the experimental investigation by assessing the % error.  
  6. The influence of materials and recommendations from it can be elaboratively discussed in the manuscript.  
  7. Conclusions should be written following the novelty, results and discussion.  
  8. The literature should be modified with recently published articles.

Reviewer 2 Report

Review Report:

Findings in the manuscript: Under the action of pressure, not only the water density change but also the axial pressure gradient exists in the fluid inside the annular gap, resulting in the plunger and plunger sleeve forming similar funnel-like shapes. Apart from this, large diameter phenomenon, high working pressure, and low fluid viscosity of the plunger pump will lead to the complicated flow of the annular channel. Moreover, considering the deformation gradient and eccentricity of the plunger pair and the compressibility of the water, the deformation equations and leakage equations of the annular channel under the laminar and turbulent flow state are derived in this study. The leakage of the annular channel under different pressure conditions is measured using a built-sealing test bench. Also, the authors elaborated on the discrepancy between the calculated model and the experimental results.

Overall, the manuscript is well-written and managed. Everything regarding the results and observations is presented significantly. Only the authors can update the literature survey by including the latest articles, such as https://doi.org/10.1002/htj.22763, https://doi.org/10.1142/S0217979222502101.

Author Response

Point 1: Findings in the manuscript: Under the action of pressure, not only the water density change but also the axial pressure gradient exists in the fluid inside the annular gap, resulting in the plunger and plunger sleeve forming similar funnel-like shapes. Apart from this, large diameter phenomenon, high working pressure, and low fluid viscosity of the plunger pump will lead to the complicated flow of the annular channel. Moreover, considering the deformation gradient and eccentricity of the plunger pair and the compressibility of the water, the deformation equations and leakage equations of the annular channel under the laminar and turbulent flow state are derived in this study. The leakage of the annular channel under different pressure conditions is measured using a built-sealing test bench. Also, the authors elaborated on the discrepancy between the calculated model and the experimental results.

Overall, the manuscript is well-written and managed. Everything regarding the results and observations is presented significantly. Only the authors can update the literature survey by including the latest articles, such as https://doi.org/10.1002/htj.22763, https://doi.org/10.1142/S0217979222502101.

Response 1:

Thanks for your valuable suggestions. We have read and cited the literature. This literature studies the heat transfer characteristics of fluid flow, which provides a reference for the follow-up research direction of my manuscript.

Reviewer 3 Report

The paper is in the scope of the Lubricants Journal. It is devoted to creating a theory for calculating the leakage flow characteristics of a plunger pump seal under laminar and turbulent flow regimes and studying the influence of eccentricity, structural deformations, and water compressibility. The paper has the following comments and questions:

1. Explain the phenomenon why the different influence of the elastic modulus and Poisson's ratio of different materials for the plunger and sleeve?

2. Explain the phenomenon why the effect of eccentricity on the leakage is so significant and why the difference of this effect is almost two times less for turbulent flow compared to laminar flow?

3. In the conclusion, explain why your work's scientific results are preferable to previous researchers' scientific results. Why didn't you use the CFD and FSI numerical methods for your studies?

4. Why were leakages in laminar and turbulent regimes compared with experiment and with each other if it was simply possible to determine the flow regime at the corresponding inlet pressure by the Reynolds number (Figure 10)?

5. Why was it impossible to change the experiment's eccentricity value in a wide range?

6. Correct the spelling and typos in the text of the paper. For example, in the paper's title, you may need to remove the pretext "on".

In the text and references, the second word in the authors' surname "Kyritsi-Yiallourou" should be capitalized.

On page 4, line 139, it is necessary to indicate a large inlet and a small outlet clearances.

On page 5, line 157, it might be better to use "velocity" instead of "speed".

Increase the size of symbols in formulas (10) and (11).

On page 8, line 208, it is necessary to delete the word "octonion".

On page 12, line 281, it is necessary to write "test bench".

On Figure 13 it is necessary to specify the "L" unit.

In the Nomenclature, the descriptions of all symbols must be in lower case.

Round 2

Reviewer 1 Report

The authors have modified the manuscript with discussed suggestions properly. hence, the manuscript can be accepted in its current form. 

Reviewer 3 Report

The paper may be published in the Lubricants journal in its revised form.